# Protocol: The role of defunctioning stoma prior to neoadjuvant therapy for locally advanced colonic and rectal cancer-A systematic review

**Mina Mesri** [1,2]*, **Louise Hitchman** [1,3], **Marina Yiaesemidou** [1,2], **Aaron Quyn** [4,5], **David Jayne** [4,5], **Ian Chetter** [1,3]

1 University of Hull, Hull, United Kingdom, 2 Bradford Teaching Hospitals NHS Foundation Trust, Bradford, United Kingdom, 3 Hull University Teaching Hospitals NHS Trust, Hull, United Kingdom, 4 University of Leeds, Leeds, United Kingdom, 5 Leeds Teaching Hospitals NHS Trust, Leeds, United Kingdom

* minamesri1@gmail.com

**Data Availability Statement:** No datasets were generated or analysed during the current study. All

## Abstract

Defunctioning stomas (ileostomy and colostomy) may be used prior to commencement of neoadjuvant therapy in patients with locally advanced colon or rectal cancer, in order to prevent clinical large bowel obstruction caused by radiotherapy associated oedema or progression of disease in patients who are not obstructed. However, the exact rate of clinical obstruction in patients undergoing neoadjuvant therapy who do not receive a defunctioning stoma is not known. Furthermore, it is not clear which factors predispose patients to developing clinical large bowel obstruction. Given that defunctioning stomas are associated with post operative and intra-operative risks, it is not currently possible to tailor defunctioning stomas to patients who have the greatest risk of developing obstruction. This systematic review which is in accordance with the Preferred Reporting Items for Systematic Reviews and Meta-Analysis statement (PRISMA), aims to define the role of defunctioning stomas in prevention of obstruction patients with locally advanced colon or rectal cancer while undergoing neoadjuvant therapy. Two researchers will perform the literature search which will include all published and "in process" articles published in the English language between 2002–2022 in the following databases: EMBASE (OVID), MEDLINE (EBSCO), CINHAL complete, Web of Science, Cochrane Central Registry of Controlled Trials, Clinical Trials Registry. The full text of the selected articles will be independently screened by two researchers against the inclusion criteria. Data will be extracted from each article regarding: study design, participants, type of intervention and outcomes. The effect size will be expressed in incidence rates and when appropriate in relative risk with 95% confidence intervals. If possible, we will perform a meta-analysis. Heterogeneity will be assessed using $I^2$ statistics. We will pool the data extracted from the randomised controlled trials to perform a meta-analysis using the Review Manager 5 software (RevMan 5). The Grades of Recommendation, Assessment, Development and Evaluation (GRADE) system will be used to assess the certainty of the evidence.

relevant data from this study will be made available upon study completion.

**Funding:** The author(s) received no specific funding for this work.

**Competing interests:** The authors have declared that no competing interests exist.

## Introduction

### Colorectal cancer

With 1.9 million new diagnoses in 2020 and 935,000 deaths, colorectal cancer accounted for the third most diagnosed cancer as well as the second most common cause of cancer deaths worldwide [1]. By 2030, it is expected that the global incidence will increase by 60% and colorectal cancer will be responsible for 1.1 million deaths [2].

Large bowel obstruction, is the initial presentation in up to 27% of patients with colorectal cancer and is more common in the elderly population [3]. Large bowel obstruction can cause bowel perforation, sepsis and death, and the prognosis of patients presenting with obstructing colorectal cancer is generally considered poor [4]. In one study of 1004 patients presenting with malignant large bowel obstruction, in-hospital mortality was found to be 12.7%, while the median survival was just 2.5 months [5].

### Treatment options for locally advanced colorectal cancer

**Surgery.** Treatment in the acute setting as a result of clinical obstruction differs greatly to the management of colorectal cancer in an elective setting in the absence of obstruction. As the former is often performed in the emergency setting in patients with poor physiological and nutritional reserve as well as haemodynamic instability, treatment focuses mainly on relief of obstruction through means of emergency "damage-control" surgery, defunctioning stoma or more recently, insertion of self-expanding endoscopic metallic stents (SEMS) with the view of performing resectional surgery in the future [6, 7]. While surgery also represents the mainstay of treatment in the absence of obstruction in the elective setting, here it aims to achieve a complete resection of major vascular pedicles and lymphatics supplying the tumour, as well as a disease-free margin and en-bloc removal of any structure adherent to the tumours in patients that have been physiologically and nutritionally built up. A defunctioning stoma may also be used in some instances in the elective setting to protect a distal anastomosis.

Following resection of rectal cancer with curative intent, the rate of local recurrence has been reported to vary from 3.7–13%, representing the second most common type of disease progression in rectal cancer, after distant metastasis [8–12]. Therefore, its rate reduction has been the target of new surgical techniques. It is now widely accepted that achieving an adequate resection margin in treatment of colon cancer and total mesorectal excision (TME) in treatment of rectal cancer improves local recurrence rate [13].

**Neo-adjuvant chemoradiotherapy.** Over the past decade, neoadjuvant treatments, including a combination of both radiotherapy and chemotherapy, have been implemented prior to definitive surgical intervention in some patients, with the primary aim of reducing the local rate of recurrence.

For rectal cancer, the European Society of Medical Oncology (ESMO) advises the use of neoadjuvant therapy for specific patients that have locally-advanced cancer with one or more poor prognostic factors including >T3c, >N1, involved extramural vascular invasion (EMVI +ve) and involved circumferential resection margin (CRM +ve) [14, 15]. There are two established neoadjuvant treatment options in locally advanced rectal cancer: Long course chemoradiotherapy (LCCRT) and short course radiotherapy (SCRT). The former involves the administration of a total of 45 Gy delivered by daily 2.8–2 Gy over the course of 5 weeks with concomitant 5-FU based chemotherapy followed by planned resection in 6–8 weeks after completion. The latter includes the delivery of 5x 5 Gy over the course of 1 week with planned surgery in the following week [16].

LCCRT and SCRT have largely demonstrated similar results when assessing the rates of 5-year distant recurrence (27–30%),5-year survival (70–74%)and late adverse grade 3–4

toxicity (5.8–8.2%) [17]. The comparable efficacy of SCRT and reduced toxicity means that it is considered an equally effective option in treatment of locally advanced rectal cancer in the frail patient population in whom chemotherapy associated toxicity may pose further delays to the start date of surgery [16]. Recently, the effect of differing intervals between neoadjuvant therapy and surgery and its effect on oncological outcomes and complication profile has been subject of ongoing research. In comparison to a straight-to-surgery approach, a delay of 4–8 weeks between SCRT and surgery has been found to have equal oncological outcomes whilst significantly reducing post-operative complication rates [18]. This latter effect is thought to be a result of the delay allowing the opportunity for patients to regain full immunity following neoadjuvant therapy, build fitness, and optimise their nutritional status prior to undergoing major abdominal surgery [18].

Magnetic resonance imaging (MRI) is an accurate tool for detecting these high risk features and consequently identifying those cancers that would benefit from downstaging prior to a safe surgical excision [19]. For the treatment of this cohort, the benefits of pre-operative radiotherapy or combination chemoradiotherapy have been well-established since the early 2000s [20, 21]. Several multi-centre trials have since demonstrated its effect in reducing the local recurrence rate as well improving survival [22–26], and most guidelines worldwide now recommend its use for downstaging prior to surgery [27–29]. More recent trials have focused on investigating different forms of neoadjuvant chemoradiotherapy (short course vs long course or total neoadjuvant therapy vs standard treatment) [18].

For colonic cancer, the typical treatment is early surgical resection followed by adjuvant chemotherapy, while the role of neoadjuvant chemotherapy is a relatively new topic. In a recent meta-analysis published in 2021, neoadjuvant chemotherapy was found to be associated with a greater rate of margin negative resection rate and improved survival [30].

In the UK, the use of neoadjuvant chemotherapy is also recommended in resectable T4 colonic tumours. This recommendation is based on the preliminary results from the unpublished FOxTROT study which demonstrated that 6 weeks of pre-operative OxFp chemotherapy for radiologically staged locally advanced operable colon cancer was associated with a substantial down-staging, tumour regression and complete clinical response [31].

Radio-chemotherapeutic agents are associated with severe adverse effects that should be considered alongside patient's fitness prior to commencement of therapy. As local tumour oedema is a common complication of radiotherapy, there is an increased risk of obstruction in rectal cancer patients receiving neoadjuvant radiotherapy [32]. Furthermore, colon cancer patients receiving neoadjuvant chemotherapy may have an increased risk of obstruction due to the delay in receiving definitive surgery and the potential for disease progression over time in spite of the treatment; however, this risk is poorly understood.

## The role of defunctioning stoma prior to neoadjuvant therapy

To prevent the risk of clinical obstruction caused by radiotherapy-related tumour oedema or disease progression over time, many surgeons opt to perform a defunctioning stoma for patients undergoing neoadjuvant chemoradiotherapy in the presence of radiological or endoscopic features of obstruction or near-obstruction, even if they are not clinically obstructed [33]. However, the exact rate at which defunctioning stoma is performed prior to commencement of neoadjuvant therapy in colorectal cancer is poorly investigated.

The advantages of this technique include eliminating the risk of bowel obstruction in the interim period between neoadjuvant chemoradiotherapy and definitive surgery, and reducing the risk of debilitating diarrhoea which may be caused secondary to therapy or the tumour itself, and therefore improving the patient's quality of life. Progression to complete bowel

obstruction may interrupt neoadjuvant chemoradiotherapy and therefore adversely affect the long-term oncological outcomes [10, 34].

The concerns with creation of a prophylactic stoma are related to disease progression caused by the delay in commencement of neoadjuvant therapy or definitive surgery while the patient recovers from major abdominal surgery. Defunctioning stomas are also associated with a number of complications which can further lengthen this interim period. With an incidence of 50%, parastomal hernias are the most common type of complication experienced by patients with any type of stoma. While some are asymptomatic, up to 70% of all patients with parastomal hernia require surgery at some point in their life due to discomfort, cosmetic dissatisfaction or more rarely bowel incarceration and strangulation [35].

Another potential problem with defunctioning stomas is their lack of or delayed reversal. A study assessing the reversal rate following defunctioning stomas created prior to low anterior resection found that most patients waited longer than 4 months post-operatively to receive a reversal and 21% of patients still had a stoma at the end of the follow-up period [36].

Due to the potential complications of defunctioning stoma, it is important that it is only offered to patients who are at high risk of developing large bowel obstruction. Currently, however, this is not possible, as the patient and disease-specific factors associated with higher risk of obstruction are unknown, and the evidence surrounding this topic is scarce. In the absence of evidence-based guidelines, the decision to offer patients a defunctioning stoma prior to neoadjuvant therapy is currently based on the presence of worrying endoscopic or radiological findings, as well as surgeon and patient preferences.

## Materials and methods

### Aims and objectives

The main aim of this systematic review is to characterise the current role of defunctioning stoma prior to neoadjuvant chemoradiotherapy (prophylactic stomas) in patients with locally advanced colorectal cancer. The main objectives of this systematic review are to:

1. Calculate the proportion of patients with colorectal cancer who undergo a prophylactic defunctioning stoma prior to neoadjuvant therapy

2. Calculate the rate of obstruction in patients undergoing neoadjuvant chemoradiotherapy

3. Determine whether or not prophylactic defunctioning stomas are associated with a delay to the commencement of neoadjuvant therapy

4. Determine whether or not prophylactic defunctioning stomas are associated with reduced rates of completion of neoadjuvant therapy

5. Characterise the complication profile of prophylactic stomas

6. Compare the long-term oncological outcomes of patients undergoing prophylactic defunctioning stoma vs patients who proceed directly to neoadjuvant therapy

### Inclusion criteria (PICO)

**Population (P).**    The study participants will include adults (>19 years of age) presenting for the first time with >T3 colorectal cancer included in the MDT process, who are awaiting any form of neoadjuvant therapy (chemotherapy/radiotherapy or a combination of both) prior to definitive surgical intervention.

This study will also exclude any adults who have previously undergone abdominal surgery for colorectal malignancy. However, abdominal surgery performed for treatment of benign

pathology will not result in exclusion. Similarly, patients have previously undergone chemora-diotherapy for primary colorectal cancer will be excluded from this study.

Furthermore, only patients with a primary colorectal cancer of adenocarcinoma origin will be included in this study and any with other types of cancer such as anal cancer. Only patients that will be undergoing surgery with curative intent at a future date will be included in this study and all patients undergoing non-surgical or surgical procedures for palliation purposes only will be excluded.

**Intervention (I).**   The intervention group will include patients who have undergone a defunctioning stoma prior to administration of neoadjuvant therapy. The included subgroups are as listed below:

Types of stoma:

All types will be included: end/ loop ileostomy as well as end/loop colostomy without any bowel resection

Neoadjuvant therapy:

1. Any type of neoadjuvant radiotherapy/chemoradiotherapy prior to definitive surgery for rectal cancer

2. Any type of neoadjuvant chemotherapy prior to definitive surgery for colon cancer

**Comparator (C).**   The comparator will be all patients that proceed directly to neoadjuvant therapy without a defunctioning stoma. The included subgroups are as listed below:

Neoadjuvant therapy:

1. Any type of neoadjuvant radiotherapy/chemoradiotherapy prior to definitive surgery for rectal cancer

2. Any type of neoadjuvant chemotherapy prior to definitive surgery for colon cancer

**Outcomes (O).**

1. Proportion of patients who undergo emergency defunctioning stoma (in the comparator population who did not receive a prophylactic defunctioning stoma)

2. Time to treatment in each group (Time taken for patients to start neoadjuvant therapy from the time of diagnosis)

3. Incidence of completion of neoadjuvant therapy in each group (This rate considers complete termination of neoadjuvant therapy or a change in the duration of therapy, i.e. if a planned LCCRT was changed into SCRT)

4. Oncological outcomes in each group (5-year survival rate, 5-year local recurrence rate and 12-month all cause mortality rate)

5. Stoma-associated complication rates (incidence of stoma prolapse, retraction and high-output stomas) in patients in the intervention group receiving a prophylactic defunctioning stoma

## Search strategy

To appraise the quality of articles and assess their eligibility for the inclusion to this review, a systematic strategy will be used. This will comprise searching databases followed by scanning the reference lists for any studies accepted for inclusion and grey literature.

Electronic searching of five databases will be performed:
Searched databases are as follows:

- EMBASE (OVID)

- MEDLINE (EBSCO)

- CINHAL complete

- Web of Science

- Cochrane Central Registry of Controlled Trials

- Clinical Trials Registry

The search terms were generated following discussion with a senior librarian at Bradford Teaching Hospital Foundation Trust and wider teams of authors including two colorectal consultants. The search terms are outlined in Table 1 in S1 Appendix. Truncation and proximity operators will be applied as necessary to broaden the search.

All published or "in-process" prospective observational studies and trials (randomised and non-randomised controlled trials), written in the English language and published in the last 20 years will be included in this systematic review. Phase I and II clinical trials and retrospective observational studies will be excluded from this review.

All the above filters will be applied to the aforementioned databases as per the inclusion criteria. If the time between the date of the search and the publication data exceeds 12 months, a second search will be carried out so that any additional studies are taken into consideration. The remaining publications will then be exported to ENDNOTE and combined so that any duplicates are removed. These will then be screened as per the eligibility criteria.

Another strategy will consist of searching through the reference list of articles that may have been missed by electronic database searches. Studies of interest will have their titles and abstracts analysed and screened as per our inclusion and exclusion criteria.

Finally, we will further augment our search by searching for grey literature. We will do this by entering our key terms into the Google internet search engine and Google Scholar search application and assessing the first 100 results. In a similar manner, we will also search Open-Grey a repository for grey literature.

## Study selection

Titles, abstracts and full texts will be screened independently by two authors using an eligibility proforma. If the two authors disagree over the eligibility of a study, this will be resolved through discussion between the two authors, and if necessary, with the wider research team.

The process of study selection will be demonstrated through a PRISMA diagram. Following full-text assessment, all excluded studies will be listed in a table, stating the reason for exclusion.

## Data extraction and management

Data extraction will be carried out by two independent researchers using pre-piloted forms. Comparisons of the extracted data will be made and any disagreements will be discussed with a third reviewer.

We will extract data on the following:

- Study design: type of study; timing of study; number of participants; length of follow-up, loss to follow up rates

- Participants: patient demographic, location and staging of colorectal cancer on presentation, traversability of tumour during the diagnostic endoscopy

- Intervention:

  - Defunctioning stoma: type of stoma (loop ileostomy vs loop colostomy), approach (laparoscopic vs open), time to definitive surgery

  - Neoadjuvant therapy: Type of therapy (i.e consolidation vs induction vs total neoadjuvant therapy), Length of therapy, types chemotherapeutic agents used, rate of acute obstruction in patients undergoing neoadjuvant therapy who did not receive a prophylactic defunctioning stoma

  - Resectional surgery: Type of surgery, presence of primary anastomosis, surgical approach (open, laparoscopic, laparoscopic-assisted and robotic), size of tumour, R0 resection, CRM, EMV, number of retrieved nodes, length of resected bowel, pathology confirmed tumour stage, need for adjuvant chemotherapy

- • Outcomes: as described above

Where the above information is not reported in retrieved articles, we will attempt to obtain this by contacting study authors.

## Assessment of risk of bias in included studies

The risk of bias will be assessed by two authors using the criteria outlined by the Risk Of Bias In Non-randomized Studies of Interventions (ROBINS-I) tool as listed by the Cochrane Handbook for systematic Reviews of Interventions when assessing the risk in observational studies and non randomised controlled studies. When assessing the risk of bias in randomised controlled trials, we will use the Cochrane risk of bias tool for RCTs (RoB-2) [37]. Any discrepancies in the calculation of the risk of bias will be resolved by discussion amongst two of the authors.

## Data analysis

Incidence of obstruction will be a mean value with 95% CI.

We will compare complication rates and oncological outcomes in intervention and control data using relative risk with 95% confidence intervals expressed in forest plots. Where feasible, we will perform a meta-analysis. We will inspect the forest plots for overlapping confidence intervals. Survival data will be expressed in percentages with 95% confidence intervals.

## Assessment of heterogeneity

Heterogeneity in each meta-analysis will be measured using the $I^2$ statistic and $Chi^2$ test values. We will consider $I^2$ value of greater than 75% high degree of heterogeneity. Values between 50%-75% will be regarded as moderate heterogeneity, while values lower than 50% will be considered as low degree heterogeneity. $I^2$ values below 25% will be signify negligible level of heterogeneity.

$Chi^2$ statistics will determine the level of significance of the calculated level of heterogeneity. The data will be considered as demonstrating highly significant level of heterogeneity if the P value is greater than 0.01 [37].

## Assessment of reporting biases

Publication bias will be considered in meta-analyses which include more than 10 trials. We will use funnel plots to visualise the standard error of each study effect size again the log of the effect size. Publication bias will be detected in the presence of asymmetry in the triangular shape of the plot [37, 38]. If publication bias is detected through asymmetry of the funnel plot, Egger's test will be used to assess the significance of this publication bias [39].

## Data synthesis

We will pool the data extracted from the randomised controlled trials to perform a meta-analysis using the Review Manager 5 software (RevMan 5). Non-randomised studies will be pooled and meta-analysed if there are considered relatively bias free and homogenous. Where available, we will use an unadjusted effect estimate for randomised trials and adjusted effect estimates for non-randomised trials and observational studies to account for potential biases.

If heterogeneity is considered high or moderate, we will use a random effects model to meta-analyse the data. If the heterogeneity is low or insignificant, we will use the fixed-effect model to calculate the effect size [37].

The Grades of Recommendation, Assessment, Development and Evaluation (GRADE) system will be used to assess the certainty of the evidence. Studies may be downgraded depending on the presence of certain limiting factors as outlines by GRADE recommendations. These include: risk of bias, imprecision, inconsistency, indirectness, and publication bias [40]. The certainty of evidence will be categorised into the following categories:

- High: There is a high level of confidence in the effect estimate and its closeness to the true effect

- Moderate: There is a moderate degree of confidence in the effect size and its closeness to the true effect

- Low: There is a limited degree of confidence in the effect estimate and the true effect may be significantly different to the effect estimate

- Very low: There is a significantly low degree of confidence in the effect estimate and it is likely that the true effect is very different to the estimate.

## Sensitivity analysis

We will carry out a sensitivity analysis to determine the extent to which our results are affected by potential bias introduced by including problematic studies at high risk of bias. If the exclusion of such studies affects the overall effect estimate significantly, they will be excluded from the meta-analysis.

## Ethical considerations

As this is a systematic review of already published literature, it will not include any participants or patients and therefore does not require an ethical approval.

## Timeline

The review will formally start on 17th May 2022 with the commencement of literature search and title and abstract screening. The review will be completed and ready for submission on 9th May 2023. The review has been registered with PROSPERO on 10/05/2022 (CRD42022331706).

## Discussion

Currently, it is not clear whether creation of a defunctioning stoma prior to neoadjuvant therapy in locally advanced colonic or rectal cancer is of any benefit to the patient. By understanding the risk profile of defunctioning stomas as well as risk of obstruction in patients who proceed to neoadjuvant therapy without a stoma, we will be able to define whether defunctioning stomas have a role in improving patient outcomes. This review will help us identify patients who are at a higher risk of clinical obstruction while undergoing neoadjuvant therapy as well as any associated delay to surgery. This together with characterising the complications associated with defunctioning stomas, will allow policy-makers to conditionally recommend this treatment based on patient and treatment characteristics, so that patients with the greatest risk of obstruction can benefit, while those with reduced risk are not unnecessarily exposed to the debilitating complications of a stoma.

## Supporting information

**S1 File. PRISMA table [41].**
(DOCX)

**S1 Appendix.**
(DOCX)

## Author Contributions

**Conceptualization:** Mina Mesri, Louise Hitchman, Marina Yiaesemidou, Aaron Quyn, David Jayne, Ian Chetter.

**Data curation:** Mina Mesri, Louise Hitchman.

**Formal analysis:** Mina Mesri.

**Investigation:** Mina Mesri.

**Methodology:** Mina Mesri.

**Project administration:** Ian Chetter.

**Software:** Mina Mesri.

**Supervision:** Marina Yiaesemidou, Aaron Quyn, David Jayne, Ian Chetter.

**Visualization:** Mina Mesri.

**Writing – original draft:** Mina Mesri.

**Writing – review & editing:** Mina Mesri, Louise Hitchman.

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
