## [Decision Letter · Decision Letter 0]

25 Jul 2022

PONE-D-22-14349Protocol: The role of defunctioning stoma prior to neoadjuvant therapy for locally advanced colonic and rectal cancer- A systematic reviewPLOS ONE

Dear Dr. Mesri,

Thank you for submitting your manuscript to PLOS ONE. After careful consideration, we feel that it has merit but does not fully meet PLOS ONE’s publication criteria as it currently stands. Therefore, we invite you to submit a revised version of the manuscript that addresses the points raised during the review process.

ACADEMIC EDITOR: The reviewers have raised a number of points which we believe major modifications are necessary to improve the manuscript, taking into account the reviewers' remarks. Please consider and address each of the comments raised by the reviewers before resubmitting the manuscript. This letter should not be construed as implying acceptance, as a revised version will be subject to re-review.

We look forward to receiving your revised manuscript.

Kind regards,

Wisit Cheungpasitporn, MD

Academic Editor

PLOS ONE

Journal Requirements: 

2. Please ensure that you include a title page within your main document. You should list all authors and all affiliations as per our author instructions and clearly indicate the corresponding author.

4. We note that this manuscript is a systematic review or meta-analysis; our author guidelines therefore require that you use PRISMA guidance to help improve reporting quality of this type of study. Please upload copies of the completed PRISMA checklist as Supporting Information with a file name “PRISMA checklist”.

Reviewers' comments:

Reviewer's Responses to Questions

**Comments to the Author**

1. Does the manuscript provide a valid rationale for the proposed study, with clearly identified and justified research questions?

Reviewer #1: Yes

Reviewer #2: Yes

2. Is the protocol technically sound and planned in a manner that will lead to a meaningful outcome and allow testing the stated hypotheses?

Reviewer #1: Yes

Reviewer #2: Yes

3. Is the methodology feasible and described in sufficient detail to allow the work to be replicable?

Reviewer #1: Yes

Reviewer #2: Yes

4. Have the authors described where all data underlying the findings will be made available when the study is complete?

Reviewer #1: Yes

Reviewer #2: No

5. Is the manuscript presented in an intelligible fashion and written in standard English?

Reviewer #1: Yes

Reviewer #2: Yes

6. Review Comments to the Author

You may also provide optional suggestions and comments to authors that they might find helpful in planning their study.

Reviewer #1: Manuscript ID: PONE-D-22-14349

The authors present a protocol: The role of defunctioning stoma prior to neoadjuvant therapy for locally advanced colonic and rectal cancer- A systematic review. They presented an interesting manuscript. However, there are some concerns that need to be addressed.

1.The authors state “I² values below 25% will be signify negligible level of heterogeneity”. Please provide citation for this paragraph.

2. The authors state “Chi² statistics will determine the level of significance of the calculated level of heterogeneity. The data will be considered as demonstrating highly significant level of heterogeneity if the P value is greater than 0.01”. Please provide citation for this paragraph.

3. The publication bias evaluation may incomplete. The authors should demonstrate both statistics and visualization.

4. The authors stated that the review will formally start on 17th May 2022 with the commencement of literature search and title and abstract screening. The review will be completed and ready for submission on 9th May 2023. Why the review starts before this protocol publish?

5. How were the search terms defined? Is there a pre-test to define the search strategy used in each database? Did the search strategy the same in all databases? Search terms in PubMed and Embase are different. The authors should attach syntax used in each database in supplementary.

Reviewer #2: This study protocol focuses in specific area. Current data is quite not enough to know all about colorectal cancer treatment. For the detail in this protocol, there are many strong points as following.

- Cleary reproducible method.

- Good management in grey literature.

- Description in all excluded studies.

However, some aspects may have to be discussed.

Major

- For population (P) in exclusion criteria, do participants who had previous abdominal surgery beside colorectal cancer need to be excluded?

- On the line 177, I am not sure that “other types of cancer” is for other gastrointestinal cancer or all of the cancer.

Minor

- On the line 24, does “assicated” mean “associated”?

- On the line 90-91, There are similar results of complications between LCCRT and SCRT. Can authors describe how large of these complications are?

- On the line 94-96, It would be nice if authors add some explanation why a delay of 4-8 weeks in surgery significantly reduce post-operative complication rate.

7. PLOS authors have the option to publish the peer review history of their article (what does this mean?). If published, this will include your full peer review and any attached files.

Reviewer #1: **Yes: **Wisit Kaewput

Reviewer #2: No

---

## [Author Response · Author response to Decision Letter 0]

9 Aug 2022

Reviewer 1

1.The authors state “I² values below 25% will be signify negligible level of heterogeneity”. Please provide citation for this paragraph.

I have used the guidance provided by Cochrane Handbook for Systematic Reviews of Interventions (2nd edition) to define the significance of heterogeneity in our review. This was cited at the beginning risk of bias section but I have now also cited this handbook throughout my methodology. Thanks for bringing this to my attention. 

2. The authors state “Chi² statistics will determine the level of significance of the calculated level of heterogeneity. The data will be considered as demonstrating highly significant level of heterogeneity if the P value is greater than 0.01”. Please provide citation for this paragraph.

I have again referred to Cochrane Handbook for Systematic Reviews of Interventions (2nd edition) to assess the significance of heterogeneity. I have now made this citation clearer in the text throughout the methodology section. 

3. The publication bias evaluation may incomplete. The authors should demonstrate both statistics and visualization.

Many thanks for your comment. In line with Cochrane Handbook for Systematic Reviews, I have now modified this section as below to demonstrate that if more than 10 studies are included in the meta-analysis, as well as a visualisation technique (funnel plot) we will also use a statistical test (Egger’s test) if asymmetry is observed:

“Publication bias will be considered in meta-analyses which include more than 10 trials. We will use funnel plots to visualise the standard error of each study effect size again the log of the effect size. Publication bias will be detected in the presence of asymmetry in the triangular shape of the plot [37, 38]. If publication bias is detected through asymmetry of the funnel plot, Egger’s test will be used to assess the significance of this publication bias [39].”

4. The authors stated that the review will formally start on 17th May 2022 with the commencement of literature search and title and abstract screening. The review will be completed and ready for submission on 9th May 2023. Why the review starts before this protocol publish?

Many thanks for your comment. Our protocol was registered with PROSPERO on 10th May 2022, a week prior to start of title and abstract screening. The protocol is unchanged since its registration on PROSPERO and therefore it is not inappropriate to begin the literature search before the protocol’s publication in PLOS ONE. 

We also note that PROSPERO’s “Guidance notes for registering a systematic review” state that, while ideally reviews should be registered with PROSPERO before screening against eligibility criteria commences (as we have done), they do accept review protocols for registration at more advanced stages, as long as they have not progressed beyond the completion of data extraction. 

5. How were the search terms defined? Is there a pre-test to define the search strategy used in each database? Did the search strategy the same in all databases? Search terms in PubMed and Embase are different. The authors should attach syntax used in each database in supplementary.

Many thanks for this comment. The search terms were generated following discussion with a senior librarian at our research institute as well as the wider team of authors including two colorectal consultants. In response to this comment, we have added the following to the Search Strategy section of the methods: “The search terms were generated following discussion with a senior librarian at Bradford Teaching Hospital Foundation Trust and wider teams of authors including two colorectal consultants. The search terms are outlined in Table 1, Appendix 1.”

The search strategy and syntax used was the same in all databases. Had we used MESH terms, the syntax for the search strategy in each database may have been different. However, we did not do so. The reason for this is that a preliminary search using the MESH terms relating to colorectal cancer AND neoadjuvant therapy in the title/abstract generated over 10,000 studies from Medline alone. We screened the first 100 of these papers and found that the papers were very loosely related to the topic of interest, especially as we are primarily investigating such a niche topic- ie. the role of defunctioning stomas prior to neoadjuvant therapy. This search strategy was therefore thought to be too non-specific. 

Following further discussion, we decided to narrow this search down by performing the search limited to the exact search terms (as provided in Appendix 1) and the titles of research papers only. This strategy was appropriate in all included databases without requiring modification. 

Using the current search strategy, we have still been able to pool a large number of studies, including 1973 papers from Embase and 1649 from Medline.

Following this comment, I have added the below note to Appendix 1 to make these points clear “The exact terms stated below were used across all search databases”

Reviewer 2

 Major revisions:

1) For population (P) in exclusion criteria, do participants who had previous abdominal surgery beside colorectal cancer need to be excluded?

Patients with previous abdominal surgery will not be excluded from this study unless they have undergone surgery for treatment of colorectal malignancy. I have now made this clear in the population (P) section.

2) On the line 177, I am not sure that “other types of cancer” is for other gastrointestinal cancer or all of the cancer.

I agree that this was not made very clear. I have now made this explicit and have also added that we will be excluding any type of cancer which is not a primary adenocarcinoma of the colon or rectum as this would not answer our primary research question. 

Minor revisions:

1) On the line 24, does “assicated” mean “associated”?

Yes, I do mean associated. Thank you for bringing this to my attention. 

2) On the line 90-91, There are similar results of complications between LCCRT and SCRT. Can authors describe how large of these complications are?

Many thanks for your comment. I have now amended this section to include the specific reported ranges of 5 year distant recurrence, 5 year survival and grade 3-4 late toxicity in the cited randomised controlled trial. As described, the values were not statistically significant.

3) On the line 94-96, It would be nice if authors add some explanation why a delay of 4-8 weeks in surgery significantly reduce post-operative complication rate.

I agree and have revised this section to provide justifications as to why the post-operative complications may be lower in the delay to surgery group, as below: 

“Recently, the effect of differing intervals between neoadjuvant therapy and surgery and its effect on oncological outcomes and complication profile has been subject of ongoing research. In comparison to a straight-to-surgery approach, a delay of 4-8 weeks between SCRT and surgery has been found to have equal oncological outcomes, whilst significantly reducing post-operative complication rates [18]. This latter effect is thought to be a result of the delay allowing the opportunity for patients to regain full immunity following neoadjuvant therapy, build fitness, and optimise their nutritional status prior to undergoing major abdominal surgery [18].

---

## [Decision Letter · Decision Letter 1]

9 Sep 2022

Protocol: The role of defunctioning stoma prior to neoadjuvant therapy for locally advanced colonic and rectal cancer- A systematic review

PONE-D-22-14349R1

Dear Dr. Mina Mesri,

We’re pleased to inform you that your manuscript has been judged scientifically suitable for publication and will be formally accepted for publication once it meets all outstanding technical requirements.

Kind regards,

Wisit Cheungpasitporn, MD

Academic Editor

PLOS ONE

Additional Editor Comments:

It appears that all comments have been appropriately responded to. I have no further comments and recommend publication.

Reviewers' comments:

Reviewer's Responses to Questions

**Comments to the Author**

1. Does the manuscript provide a valid rationale for the proposed study, with clearly identified and justified research questions?

Reviewer #1: Yes

Reviewer #2: Yes

2. Is the protocol technically sound and planned in a manner that will lead to a meaningful outcome and allow testing the stated hypotheses?

Reviewer #1: Yes

Reviewer #2: Yes

3. Is the methodology feasible and described in sufficient detail to allow the work to be replicable?

Reviewer #1: Yes

Reviewer #2: Yes

4. Have the authors described where all data underlying the findings will be made available when the study is complete?

Reviewer #1: Yes

Reviewer #2: Yes

5. Is the manuscript presented in an intelligible fashion and written in standard English?

Reviewer #1: Yes

Reviewer #2: Yes

6. Review Comments to the Author

You may also provide optional suggestions and comments to authors that they might find helpful in planning their study.

Reviewer #1: The authors had adequate addressed all my previous concerns. I have no additional comments about publication ethics.

Reviewer #2: All the major and minor concerns have been revised properly.

Especially for Major concerns:

- Exclusion criteria and other types of cancer have been clarified.

7. PLOS authors have the option to publish the peer review history of their article (what does this mean?). If published, this will include your full peer review and any attached files.

Reviewer #1: **Yes: **Wisit Kaewput

Reviewer #2: No

---

## [Editor Report · Acceptance letter]

13 Sep 2022

PONE-D-22-14349R1 

Protocol: The role of defunctioning stoma prior to neoadjuvant therapy for locally advanced colonic and rectal cancer- A systematic review 

Dear Dr. Mesri:

I'm pleased to inform you that your manuscript has been deemed suitable for publication in PLOS ONE. Congratulations! Your manuscript is now with our production department. 

Kind regards, 

on behalf of

Dr. Wisit Cheungpasitporn 

Academic Editor

PLOS ONE